# Learning to Infer User Interface Attributes from Images

## Abstract

We explore a new domain of learning to infer user interface attributes that helps developers automate the process of user interface implementation. Concretely, given an input image created by a designer, we learn to infer its implementation which when rendered, looks visually the same as the input image. To achieve this, we take a black box rendering engine and a set of attributes it supports (e.g., colors, border radius, shadow or text properties), use it to generate a suitable synthetic training dataset, and then train specialized neural models to predict each of the attribute values. To improve pixel-level accuracy, we also use imitation learning to train a neural policy that refines the predicted attribute values by learning to compute the similarity of the original and rendered images in their attribute space, rather than based on the difference of pixel values. We instantiate our approach to the task of inferring Android `Button` attribute values and achieve 92.5% accuracy on a dataset consisting of real-world Google Play Store applications.

## 1 Introduction

With over 5 million applications in Google Play Store and Apple App Store and over a billion webpages, a significant amount of time can be saved by automating even small parts of their development. To achieve this, several tools have been recently developed that help user interface designers explore and quickly prototype different ideas, including `Sketch2Code` (Microsoft, 2018) and `InkToCode` (Corrado et al., 2018), which generate user interface sketches from hand-drawn images, `Swire` (Huang et al., 2019) and `Rico` (Deka et al., 2017), which allow retrieving designs similar to the one supplied by the user and `Rewire` (Swearngin et al., 2018), which transforms images into vector representations consisting of rectangles, circles and lines. At the same time, to help developers implement the design, a number of approaches have been proposed that generate *layout* code that places the user interface components at the desired position (e.g., when resizing the application). These include both symbolic synthesis approaches such as `InferUI` (Bielik et al., 2018), which encodes the problem as a satisfiability query of a first-order logic formula, as well as statistical approaches (Beltramelli, 2018; Chen et al., 2018), which use encoder-decoder neural networks to process the input image and output the corresponding implementation.

In this work, we explore a new domain of inferring an implementation of an user interface component from an image which when rendered, looks visually the same as the input image. Going from an image to a concrete implementation is a time consuming, yet necessary task, which is often outsourced to a company for a high fee (replia, 2019; psd2android, 2019; psd2mobi, 2019). Compared to prior work, we focus on the pixel-accurate implementation, rather than on producing sketches or the complementary task of synthesizing layouts that place the components at the desired positions.

Concretely, given a black box rendering engine that defines a set of categorical and numerical attributes of a component, we design a two step process which predicts the attribute values from an input image – *(i)* first, we train a neural model to predict the most likely initial attribute values, and then *(ii)* we use imitation learning to iteratively refine the attribute values to achieve pixel-level accuracy. Crucially, all our models are trained using synthetic datasets that are obtained by sampling the black box rendering engine, which makes it easy to train models for other attributes in the future. We instantiate our approach to the task of inferring the implementation of Android `Button` attributes and show that it generalizes well to a real-world dataset consisting of buttons found in existing Google Play Store applications. In particular, our approach successfully infers the correct attribute values in 94.8% and 92.5% of the cases for the synthetic and the real-world datasets, respectively.

## 2 RELATED WORK

As an application, our work is related to a number of recently developed tools in the domain of user interface design and implementation with the goal of making developers more productive, as discussed in Section 1. Here we give overview of the related research from a technical perspective.

**Inverting rendering engines to interpret images**   The most closely related work to ours phrases the task of inferring attributes from images as the more general task of learning to invert the rendering engines used to produce the images. For example, Wu et al. (2017) use reinforcement learning to train a neural pipeline that given an image of a cartoon scene or a Minecraft screenshot, identifies objects and a small number of high level features (e.g., whether the object is oriented left or right). Ganin et al. (2018) also use reinforcement learning, but with an adversarially learned reward signal, to generate a program executed by a graphics engine that draws simple CAD programs or handwritten symbols and digits. Johnson et al. (2018) and Ellis et al. (2018) design a neural architecture that generates a program that when rendered, produces that same 2D or 3D shape as in the input image. While Johnson et al. (2018) train the network using a combination of supervised pretraining and reinforcement learning with a custom reward function (using Chamfer distance to measure similarity of two objects), Ellis et al. (2018) use a two step process that first uses supervised learning to predict a set of objects in the image and then synthesizes a program (e.g., containing loops) that draws them.

In comparison to these prior works, our approach differs in three key aspects. First, the main challenge in prior works is predicting the set of objects contained in the image and how to compose them. Instead, the focus of our work is in predicting a set of object properties after the objects in the image were already identified. Second, instead of using the expensive REINFORCE (Williams, 1992) algorithm (or its variation) to train our models, we use a two step process that first pretrains the network to make an initial prediction and then uses imitation learning to refine it. This is possible because, in our setting, there is a fixed set of attributes known in advance for which we can generate a suitable synthetic dataset used by both of these steps. Finally, because our goal is to learn pixel-accurate attribute values, the refinement loop takes as input both the original image, as well as the rendered image of the current attribute predictions. As a result, we do not require our models to predict pixel-accurate rendering of an attribute value but instead, to only predict whether the attribute values in two images are the same or in which direction they should be adjusted.

**Attribute prediction**   Optical character recognition (Jaderberg et al., 2016; Lyu et al., 2018; Jaderberg et al., 2014; Gupta et al., 2016) is a well studied example of predicting an attribute from an image with a large number of real-world applications. Other examples include predicting text fonts (Zhao et al., 2018; Wang et al., 2015; Chen et al., 2014), predicting eye gaze (Shrivastava et al., 2017), face pose and lighting (Kulkarni et al., 2015), chair pose and content (Wu et al., 2018) or 3D object shapes and pose (Kundu et al., 2018), to name just a few. The attribute prediction network used in our work to predict the initial attribute value is similar to these existing approaches, except that it is applied to a new domain of inferring user interface attributes. As a result, while some of the challenges remain the same (e.g., how to effectively generate synthetic datasets), our main challenge is designing a pipeline, together with a network architecture capable of achieving pixel-level accuracies on a range of diverse attributes.

## 3 BACKGROUND: USER INTERFACE ATTRIBUTES

Visual design of user interface components can be specified in many different ways – by defining a program that draws on a canvas, by defining a program that instantiates components at runtime and manipulates their properties, declaratively by defining attribute values in a configuration file (e.g., using CSS), or by using a bitmap image that is rendered in place of the components. In our work, we follow the best practices and consider the setting where the visual design is defined declaratively, thus allowing separating the design from the logic that controls the application behaviour.

Formally, let $\mathcal{C}$ denote a component with a set of attributes $\mathcal{A}$. The domain of possible values of each attribute $a_i \in \mathcal{A}$ is denoted as $\Theta_i$. As all the attributes are rendered on a physical device, their domains are finite sets containing measurements in pixels or a set of categorical values. For example, the domain for the text color attribute is three RGB channels $\mathbb{N}^{3 \times [0,255]}$, the domain for text gravity is $\{\texttt{top}, \texttt{left}, \texttt{center}, \texttt{right}, \texttt{bottom}\}$ and the domain for border width is $\mathbb{N}^{[0,20]}$. We distinguish

two domain types: *(i)* comparable (e.g., colors, shadows or sizes) for which a valid distance metric $d\colon \Theta \times \Theta \to \mathbb{N}^{[0,\infty)}$ exists, and *(ii)* uncomparable (e.g., font types or text gravity) for which the distance between any two attribute values is equal to one. We use $\boldsymbol{\Theta} \subseteq \Theta_1 \times \cdots \times \Theta_n$ to denote the space of all possible attribute configurations, and use the function $\mathtt{render}\colon \boldsymbol{\Theta} \to \mathbb{R}^{3 \times h \times w}$ to denote an image with width $w$, height $h$ and three color channels obtained by rendering the attribute configuration $\boldsymbol{y} \in \boldsymbol{\Theta}$. Furthermore, we use the notation $\boldsymbol{y} \sim \boldsymbol{\Theta}$ to denote a random sample of attribute values from the space of all valid attribute configurations. Finally, we note that attributes often affect the same parts of the rendered image (e.g., the shadow is overlaid on top of the background) and they are in general not independent of each other (e.g, changing the border width affects the border radius).

# 4 LEARNING TO INFER USER INTERFACE ATTRIBUTES FROM IMAGES

We now present our approach for learning to infer user interface component attributes from images.

**Problem statement**   Given an input image $\mathcal{I} \in \mathbb{R}^{3 \times h \times w}$, our goal is to find an attribute configuration $\boldsymbol{y} \in \boldsymbol{\Theta}$ which when rendered, produces an image most visually similar to $\mathcal{I}$:

$$\arg\min_{\boldsymbol{y} \in \boldsymbol{\Theta}} cost(\mathcal{I}, \mathtt{render}(\boldsymbol{y}))$$

where $cost : \mathcal{I} \times \mathcal{I} \to \mathbb{R}^{[0,\infty)}$ is a function that computes the visual similarity of a given user interface component in two images. It returns zero if the component looks visually the same in both images or a positive real value denoting the degree to which the attributes are dissimilar.

The first challenge that arises from the problem statement above is how to define the *cost* function. Pixel based metrics, such as mean squared error of pixel differences, are not suitable and instead of producing images with similar attribute values, produce images that have on average similar colors. Training a discriminator also does not work, as all the generated images are produced by rendering a set of attributes and are true images by definition. Finally, the *cost* can be computed not over the rendered image but by comparing the predicted attributes $\boldsymbol{y}$ with the ground-truth labels. Unfortunately, even if we would spend the effort and annotated a large number of images with their ground-truth attributes, using a manually annotated dataset restricts the space of models that can be used to infer $\boldsymbol{y}$ to only those that do supervised learning. In what follows we address this challenge by showing how to define the *cost* function over attributes (used for supervise learning) as well as over images (used for reinforcement learning), both by using a synthetically generated dataset.

**Our approach**   To address the task of inferring user interface attributes from images, we propose a two step process that – *(i)* first selects the most likely initial attribute values $\arg\max_{\boldsymbol{y} \in \boldsymbol{\Theta}} p(\boldsymbol{y} \mid \mathcal{I})$ by learning a probability distribution of attribute values conditioned on the input image, and then *(ii)* iteratively refines the attribute values by learning a policy $\pi(\Delta \boldsymbol{y}^{(i)} \mid \mathcal{I}, \mathtt{render}(\boldsymbol{y}^{(i)}))$ that represents the probability distribution of how each attribute should be changed, conditioned on both the original image, as well as the rendered image of the attribute configuration $\boldsymbol{y}^{(i)}$ at iteration $i$. We use the policy $\pi$ to define the *cost* between two images as $cost(\mathcal{I}, \mathcal{I}') := 1 - \pi(\Delta \boldsymbol{y} = 0 \mid \mathcal{I}, \mathcal{I}')$. That is, the cost is defined as the probability that the two images are not equal in the attribute space.

We illustrate both steps in Figure 1 with an example that predicts attributes of a `Button` component. In Figure 1 *(a)*, the input image is passed to a set of convolutional neural networks, each of which is trained to predict a single attribute value. In our example, the most likely value predicted for the border width is `2dp` while the most likely color of the border is ■ `#4a4a4a`. Then, instead of returning the most likely attribute configuration $\boldsymbol{y}$, we take advantage of the fact that it can be rendered and compared to the original input image. This give us additional information that is used to refine the predictions as shown in Figure 1 *(b)*. Here, we use a pair of siamese networks (pretrained on the prediction task) to learn the probability distribution over changes required to make the component attributes in both images the same. In our example, the network predicts that the border color and the text gravity attributes have already the correct values but the border width should be decreased by `2dp` and the shadow should be increased by `4dp`. Then, due to the large number of different attributes that affect each other, instead of applying all the changes at once, we select and apply a single attribute change. In our example, the $\Delta y$ corresponds to adjusting the value of `border width` by $-2\text{dp}$. Since the change is supposed to correct a mispredicted attribute value, we accept it only if it indeed makes the model more confident that the prediction is correct.

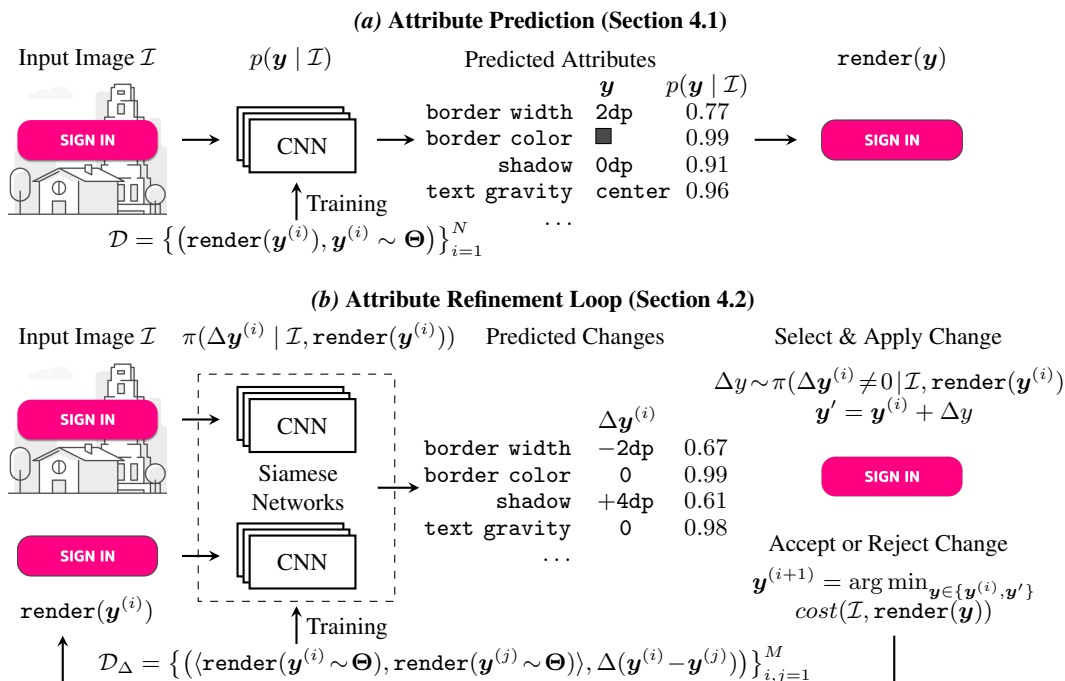

Figure 1: *(a)* Illustration of the attribute prediction network which takes an input an image with a component (a `Button`) and predicts all the component attribute values. *(b)* Refinement loop which renders the attribute values obtained from *(a)* and iteratively refines them to match the input image.

**Synthetic datasets** We instantiate our approach by training it purely using synthetic datasets, while ensuring that it generalizes well to real-world images. This allows us to avoid the expensive task of collecting and annotating real-world datasets and more importantly, makes our approach easily applicable to new domains and attributes. In particular, given a space of possible attribute configurations $\Theta$ and a rendering function `render`, we generate two different datasets $\mathcal{D}$ and $\mathcal{D}_\Delta$ used to train the attribute prediction network and the policy $\pi$, respectively. The dataset $\mathcal{D} = \{(\texttt{render}(\boldsymbol{y}^{(i)}), \boldsymbol{y}^{(i)} \sim \boldsymbol{\Theta})\}_{i=1}^{N}$ is constructed by sampling a valid attribute configuration $\boldsymbol{y}^{(i)} \sim \boldsymbol{\Theta}$ and rendering it to produce the input image. To generate $\mathcal{D}_\Delta$, we sample two attribute configurations $\boldsymbol{y}^{(i)}, \boldsymbol{y}^{(j)} \sim \boldsymbol{\Theta}$ that are used to render two input images and train the network to predict the difference of their attributes, that is, $\mathcal{D}_\Delta = \{(\langle \texttt{render}(\boldsymbol{y}^{(i)} \sim \boldsymbol{\Theta}), \texttt{render}(\boldsymbol{y}^{(j)} \sim \boldsymbol{\Theta}) \rangle, \Delta(\boldsymbol{y}^{(i)} - \boldsymbol{y}^{(j)}))\}_{i,j=1}^{M}$.

For both datasets, to avoid overfitting when training models for attributes with large domain of possible values, we sample only a subset of attributes, while setting the remaining attributes to values from the previous example $\boldsymbol{y}^{(i-1)}$. As a result, every two consecutive samples are similar to each other, since a subset of their attributes is the same. Further, because the real-world images do not contain components in isolation but together with other components that fill the rest of the screen, we introduce three additional attributes $x_{\text{pos}} \in \mathbb{N}$, $y_{\text{pos}} \in \mathbb{N}$ and `background`. We use $x_{\text{pos}}$ and $y_{\text{pos}}$ to denote the horizontal and vertical position of the component in the image, respectively. This allows the network to learn robust predictions regardless of the component position in the image. We use `background` to select the background on which the component is rendered. We experimented with three different choices of backgrounds – only while color, random solid color and overlaying the component on top of an existing application, all of which are evaluated in Section 5.

## 4.1 ATTRIBUTE PREDICTION

The attribute prediction network architecture is a multilayer convolutional neural network (CNN) followed by a set of fully connected layers. The multilayer convolutional part consists of 6 repetitive sequences of convolutional layers with ReLU activations, followed by batch normalization and a max-pooling layer of size 2 and stride 2. For the convolutional layers we use $3 \times 3$ filters of size 32, 32, 64, 64, 128 and 128, respectively. The result of the convolutional part is then flattened and

connected to a fully connected layer of size 256 with ReLU activation followed by a final softmax layer (or a single neuron for regression). We note that this is not a fixed architecture and instead, it is adapted to a given attribute by performing an architecture search, as shown in Section 5.

**Supporting multiple input sizes** To support user interface components of different sizes, we select the input image dimension such that it is large enough to contain them. This is necessary as our goal is to infer pixel-accurate attribute values and scaling down or resizing the image to a fixed dimension is not an option, as it leads to severe performance degradation. However, this is problematic, as most of the input images are smaller or cropped in order to remove other components. As a result, before feeding the image to the network we need to increase the dimension of the input without resizing or scaling. To achieve this, we pad the missing pixels with the values of edge pixels, which improves the generalization to real-world images as shown in Section 5.1 and illustrated in Appendix C.

**Optimizations** To improve the accuracy and reduce the variance of the attribute prediction network, we perform the following two optimizations. First, we compute the most likely attribute value by combining multiple predictions and selecting the most likely among them. This is achieved by generating several perturbations of the input image, each of which shifts the image randomly, horizontally by $\epsilon_x \sim \mathcal{U}(-t, t)$ and vertically by $\epsilon_y \sim \mathcal{U}(-t, t)$, where $t \in \mathbb{N}$. This is similar to ensemble models but instead of training multiple models we generate and evaluate multiple inputs.

Second, to improve the accuracy of the color attributes, we perform color clipping by picking the closest color to one of those present in the input image. To reduce the set of all possible colors, we use saliency maps (Simonyan et al., 2013) to select a subset of the pixels most relevant for the prediction. In our experiments we keep only the pixels with the normalized saliency value above 0.8. Then, we clip the predicted color to the closest color from the top five most common colors among the pixels selected by the saliency map. We provide illustration of the color clipping in Appendix D.

## 4.2 ATTRIBUTE REFINEMENT LOOP

We now describe how to learn a function $\pi(\Delta \boldsymbol{y} \mid \mathcal{I}, \texttt{render}(\boldsymbol{y}))$ that represents probability distribution of how each attribute should be changed, conditioned on both the original image as well as the rendered image of the current attribute configuration $\boldsymbol{y}$. We can think of $\pi$ as a policy, where the actions correspond to changing an attribute value and the state is a tuple of the original and the currently rendered image. We can then apply imitation learning to train the policy $\pi$ on a synthetic dataset $\mathcal{D}_\Delta = \{(\langle \texttt{render}(\boldsymbol{y}^{(i)} \sim \boldsymbol{\Theta}), \texttt{render}(\boldsymbol{y}^{(j)} \sim \boldsymbol{\Theta}) \rangle, \Delta(\boldsymbol{y}^{(i)} - \boldsymbol{y}^{(j)}))\}_{i,j=1}^M$. Because the range of possible values $\Delta(\boldsymbol{y}^{(i)} - \boldsymbol{y}^{(j)})$ can be large and sparse, we limit the range by clipping it to an interval $[-c, c]$ (where $c$ is a hyperparameter set to $c = 5$ for all the attributes in our evaluation). To fix a change larger than $c$, we perform sequence of small changes. For comparable attributes, the delta between two attribute values is defined as their distance $\Delta(\boldsymbol{y}_k^{(i)} - \boldsymbol{y}_k^{(j)}) := d(\boldsymbol{y}_k^{(i)} - \boldsymbol{y}_k^{(j)})$. For uncomparable attributes, the delta is binary and determines whether the value is correct or not.

The model architecture used to represent $\pi$ consists of two convolutional neural networks with shared weights $\theta$, also called siamese neural networks, each of which computes a latent representation of the input image $\boldsymbol{h}_x = f_\theta(\mathcal{I})$ and $\boldsymbol{h}_r = f_\theta(\texttt{render}(\boldsymbol{y}))$. The function $f_\theta$ has the same architecture as the attribute prediction network, except that we replace the fully connected layer with one that has a bigger size and remove the last softmax layer. Then, we combine the latent features $\boldsymbol{h}_x$ and $\boldsymbol{h}_r$ into a single vector $\boldsymbol{h} = [\boldsymbol{h}_x; \boldsymbol{h}_r; \boldsymbol{h}_x + \boldsymbol{h}_r; \boldsymbol{h}_x - \boldsymbol{h}_r; \boldsymbol{h}_x \odot \boldsymbol{h}_r]$, where $\odot$ denotes element-wise multiplication. Finally, the vector $\boldsymbol{h}$ is passed to a fully connected layer of size 256 with ReLU activations, followed by a final softmax layer. Once the models are trained, we perform the refinement loop as follows:

*Select attribute to change.* As in general attributes interfere with each other, in each refinement iteration we adjust only a single attribute, which is chosen by sampling from the following distribution:

$$P[A = a_i] = \frac{1 - \pi(\Delta \boldsymbol{y}_i = 0 \mid \mathcal{I}, \texttt{render}(\boldsymbol{y}))}{\sum_{k=1}^{|\mathcal{A}|} 1 - \pi(\Delta \boldsymbol{y}_k = 0 \mid \mathcal{I}, \texttt{render}(\boldsymbol{y}))}$$

where $\pi(\Delta \boldsymbol{y}_k = 0 \mid \mathcal{I}, \texttt{render}(\boldsymbol{y}))$ denotes the probability that the $k$-th attribute should *not* be changed, that is, the predicted change is zero. Since we train a separate model for each attribute, the probability that the given attribute *should* be changed is $1 - \pi(\Delta \boldsymbol{y}_k = 0 \mid \mathcal{I}, \texttt{render}(\boldsymbol{y}))$.

*Select attribute's new value.* For comparable attributes, we adjust their value by sampling from the probability distribution computed by the policy $\pi$, which contains changes in range $[-c, c]$. For uncomparable attributes another approach has to be chosen, since the delta prediction network computes only whether the attribute is correct and not how to change it. Instead, we select the new value by sampling from the probability distribution computed by the corresponding attribute prediction network.

*Accept or reject the change.* In a typical setting, we would accept the proposed changes as long as the model predicts that an attribute should be changed. However, in our domain we can render the proposed changes and check whether the result is consistent with the model. Concretely, we accept the change $y'$ if it reduces the *cost*, that is, $cost(\mathcal{I}, \texttt{render}(y')) < cost(\mathcal{I}, \texttt{render}(y))$. Note that this optimization is possible only if the change was supposed to fix the attribute value, that is, the change was in the range $(-c, c)$ or the attribute is uncomparable.

## 5 EVALUATION

To evaluate the effectiveness of our approach, we apply it to the task of generating Android `Button` implementations. Concretely, we predict the following 12 attributes – border color, border width, border radius, height, width, padding, shadow, main color, text color, text font type, text gravity and text size. We do not predict the text content for which specialized models already exist (Jaderberg et al., 2016; Lyu et al., 2018; Jaderberg et al., 2014; Gupta et al., 2016). We provide domains and the visualization of all these attributes in Appendix A. In what follows we first describe our datasets and evaluation metrics, then we present a detailed evaluation of our approach consisting of the attribute prediction network (Section 4.1) and the refinement loop (Section 4.2).

**Datasets** To train the attribute prediction network we use a synthetic dataset $\mathcal{D}$ containing $\approx$20,000 images and their corresponding attributes as described in Section 4. To train the refinement loop we use a second synthetic dataset $\mathcal{D}_\Delta$, also containing $\approx$20,000 image pairs. During training we perform two iterations of `DAgger` (Ross et al., 2011), each of which generates $\approx$20,000 samples obtained by running the policy on the initial training dataset. To evaluate our models we use two datasets – *(i)* synthetic $\mathcal{D}^{syn}$ generated in the same way as for training, and *(ii)* real-world $\mathcal{D}^{gplay}$ we obtained by manually implementing 110 buttons in existing Google Play Store applications. The illustration of samples and our inferred implementations for both datasets are provided in Appendix F.

**Evaluation metrics** To remove clutter, we introduce an uniform unit to measure attribute similarity called *perceivable difference*. We say that two attributes have the *same* (=) perceivable difference if their values are the same or almost indistinguishable. For example, the text size is perceivably the same, if the distance of the predicted $y$ and the ground-truth $y^*$ value is $d(y, y^*) \leq 1$, while the border width is perceivably the same only if it is predicted perfectly, i.e., $d(y, y^*) = 0$. The formal definition of perceivable difference with visualizations of all attributes is provided in Appendix E.

### 5.1 ATTRIBUTE PREDICTION

A detailed summary of the variations of our attribute prediction models and their effect on performance is shown in Table 1. To enable easy comparison, we selected a good performing instantiation of our models, denoted as *core*, against which all variations in the rows *(A)-(E)* can be directly compared. Based on our experiments, we then select the *best* configuration that achieves accuracy 93.6% and 91.4% on the synthetic and the real-world datasets, respectively. All models were trained for 500 epochs, with early-stopping of 15 epochs, using a batch size of 128 and initial learning rate of 0.01. In what follows we provide a short discussion of each of the variations from Table 1.

*Image background (A)* We trained our models on synthetic datasets with three different component backgrounds of increasing complexity – white color, random solid color and user interface screenshot. Unsurprisingly, the models trained with white background fail to generalize to real-world datasets and achieve only 56.7% accuracy. Perhaps surprisingly, although the models trained with the screenshot background improve significantly, they also fail to generalize and achieve only 75.6% accuracy. Upon closer inspection, this is because overlaying components over existing images often introduces small visual artefacts around the component. On the other hand, random color backgrounds generalize well to real-world dataset as they have enough variety and no visual artefacts.

Table 1: Variations of the attribute prediction network and their effect on the model accuracy.

| | Network Architecture | | | Dataset & Input Preprocessing | | | Accuracy | |
|---|---|---|---|---|---|---|---|---|
| | $arch$ | $lr_{rd}$ | $color_{clip}$ | $input_{size}$ | $tr$ | $background$ | $\mathcal{D}^{syn}_{\equiv}$ | $\mathcal{D}^{gplay}_{\equiv}$ |
| $core$ | $C_0, R_0$ | - | $saliency_{top5}$ | $150{\times}330$ | $tr_2$ | $rand$ | 92.7% | 90.1% |
| $(A)$ | | | | | | $white$ | 88.9% | 56.7% |
| | | | | | | $screenshot$ | 88.6% | 75.6% |
| $(B)$ | | | | | $center$ | | 89.4% | 82.4% |
| | | | | | $tr_1$ | | 92.3% | 90.3% |
| $(C)$ | | | | $135{\times}310$ | | | 92.5% | 91.2% |
| | | | | $180{\times}350$ | | | 92.5% | 88.7% |
| $(D)$ | | | $image_{top5}$ | | | | 89.2% | 85.9% |
| | | | $image_{all}$ | | | | 87.1% | 83.0% |
| | | | $none$ | | | | 74.1% | 72.0% |
| $(E)$ | $C_3$ | | | | | | 83.2% | 83.4% |
| | $C_6$ | | | | | | 88.6% | 88.2% |
| | $R_1$ | | | | | | 62.6% | 64.3% |
| | $R_2$ | | | | | | 67.7% | 65.3% |
| | $\dots$ | | | | | | $\dots$ | $\dots$ |
| **best** | $C_6, R_2$ | 0.1 | $saliency_{top5}$ | $135{\times}310$ | $tr_2$ | $rand$ | **93.6%** | **91.4%** |

$arch$ model architecture      $tr$ input transformation      $lr_{rd}$ reduced learning rate on plateau

*Affine transformations (B)*  Since the components can appear in any part of the input image, we use three methods to generate the training datasets – $tr_1$ places the component randomly at any position with a margin of at least 20 pixels of the image border, $tr_2$ places the component in the middle of the image with a horizontal offset $\epsilon_x \sim \mathcal{U}(-13, 13)$ and vertical offset $\epsilon_y \sim \mathcal{U}(-19, 19)$, and $center$ always places the component exactly in the center. We can see that using either $tr_1$ or $tr_2$ leads to significantly more robust model and increases the real-world accuracy by $\approx 8\%$.

*Input image size & padding (C)*  As our goal is to perform pixel-accurate predictions, we do not scale down or resize the input images. However, since large images include additional noise (e.g., parts of the application unrelated to the predicted component), we measure how robust our model is to such noise by training models for three different input sizes – $135{\times}310$, $150{\times}330$ and $180{\times}350$. While the accuracy on the synthetic dataset is not affected, the real-world accuracy shows a slight decrease for larger sizes that contain more noise. However, note that the decrease is so small because of our padding technique, which extends the component to a larger size by padding the missing pixels with edge pixel values. When using no padding, the accuracy of the real-world dataset drops to $72\%$ and when padding with a $white$ color the accuracy drops even further to $71\%$ (not shown in Table 1).

*Color clipping (D)*  We experimented with different color clipping techniques – $saliency_{top5}$ that considers the top 5 colors in the saliency map of a given attribute, $image_{top5}$ that considers the top 5 colors in the image, and $image_{all}$ that considers all the colors in the image. The color clipping using saliency maps performs the best and leads to more than $3\%$ and $16\%$ improvements over other types of clipping or using no clipping, respectively. While the other types of color clipping also perform reasonably well, they typically fail for images that include many colors, where the saliency map helps focusing only on the colors relevant to the prediction. We note that the color clipping works best for components with solid colors and the improvement for gradient colors is limited.

*Network architecture (E)*  We adapt the architecture presented in Section 4.1 for each attribute, by performing a small scale architecture search. Concretely, we choose between using classification ($C$) or regression ($R$), the kernel sizes, the number of output channels, whether we use pooling layer and whether we use additional fully connected layer before the softmax layer. Although the results in Table 1 are not directly comparable, as they provide only the aggregate accuracy over all attributes (additionally for regression experiments we consider only numerical attributes), they do show that such architectural choices have a significant impact on the network's performance.

Table 2: Accuracy of the attribute refinement loop instantiated with different similarity metrics. The accuracy shown in brackets denotes the improvement compared to the initial attribute values.

| Metric | Random Attribute Initialization | | Best Prediction Initialization | |
| | $\mathcal{D}^{syn}_{\cong}$ | $\mathcal{D}^{gplay}_{\cong}$ | $\mathcal{D}^{syn}_{\cong}$ | $\mathcal{D}^{gplay}_{\cong}$ |
| --- | --- | --- | --- | --- |
| **Our Work** | | | | |
| Learned Dst. | **94.4%** (+57.3%) | **91.3%** (+53.3%) | **94.8%** (+1.2%) | **92.5%** (+1.1%) |
| **Baselines** | | | | |
| Pixel Sim. | 59.6% (+22.5%) | 65.0% (+27.0%) | 93.6% ( 0.0%) | 91.1% (−0.3%) |
| Structural Sim. | 81.1% (+44.0%) | 71.9% (+33.9%) | 93.4% (−0.2%) | 89.3% (−2.1%) |
| Wasserstein Dst. | 63.4% (+26.3%) | 61.8% (+23.8%) | 91.8% (−1.8%) | 89.6% (−1.8%) |

## 5.2 ATTRIBUTE REFINEMENT

To evaluate our attribute refinement loop, we perform two experiments that refine the attribute values: *(i)* starting from random initial values, and *(ii)* starting from values computed by the attribute prediction network. For both experiments, we show that the refinement loop improves the accuracy of the predicted attribute values, as well as significantly outperforms other similarity metrics used as a baseline. We trained all our models for 500 epochs, with early-stopping of 15 epochs, using a batch size of 64, learning rate of 0.01 and gradient clipping of 3, which is necessary to make the training stable. Further, we initialize the siamese networks with the pretrained weights of the best attribute prediction network, which leads to both improved accuracy of 4%, as well as faster convergence when compared to training from scratch. Finally, we introduce a hyperparameter that controls which attributes are refined. This is useful as it allows the refinement loop to improve the overall accuracy even if only a subset of the attributes values can be refined successfully.

**Attribute refinement improves accuracy** The top row in Table 2 *(right)* shows the accuracy of our refinement loop when applied starting from values predicted by the best attribute prediction network. Based on our hyperparameters, we refined the following six attributes – text size, text gravity, text font, shadow, width and height. The refined attributes are additionally set to random initial values for the experiment in Table 2 *(left)*. The overall improvement for both synthetic and real-world dataset is $\approx 1.1\%$ when starting from the values predicted by the attribute prediction network. When starting from random values the refinement loop can still recover predictions of almost the same quality, although with $\approx 12\times$ more refinement iterations. The reason why the improvement is not higher is mainly because $\approx 5\%$ of the errors are performed when predicting the text color and the text padding, for which both the attribute prediction networks and the refinement loop work poorly. This suggest that a better network architecture is needed to improve the accuracy of these attributes.

**Effectiveness of our learned similarity metric** To evaluate the quality of our learned *cost* function, which computes image similarity in the attribute space, we use the following similarity metrics as a baseline – pairwise *pixel difference*, *structural similarity* (Wang et al., 2004), and *Wasserstein distance*. As the baseline metrics depend heavily on the fact that the input components are aligned in the two images (e.g., when computing pairwise pixel difference), for a fair comparison we add a manual preprocessing step that centers the components in the input image. The results from Table 2 show that all of these metrics are significantly worse compared to our learned *cost* function. Even though they provide some improvement when starting from random attributes, the improvement is limited and all of them result in accuracy decrease when used starting from good attributes.

## 6 CONCLUSION

We present an approach for learning to infer user interface attributes from images. We instantiate it to the task of learning the implementation of the Android `Button` component and achieve 92.5% accuracy on a dataset consisting of Google Play Store applications. We show that this can be achieved by training purely using suitable datasets generated synthetically. This result indicates that our method is a promising step towards automating the process of user interface implementation.

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

APPENDIX

We provide six appendices. In Appendix A we define domains of all the attributes considered in our work and include their visualizations. In Appendix B we include details of the stopping criterion used in the refinement loop and provide accuracy breakdown of the individual attribute values. In Appendix C we illustrate three different techniques to pad images to a larger size. In Appendix D we show an example of using color clipping with saliency maps. In Appendix E we formally define the *perceivable different* metric used to compute accuracy in our evaluation. Finally, in Appendix F we provide examples of images and the inferred implementations for samples in both the synthetic and real-world datasets.

## A   ANDROID BUTTON ATTRIBUTES

We provide definition of all the attributes considered in our work as well as their visualization in Table 3. For border radius we use a special value $\infty$ to denote round buttons.

Table 3: `Button` attribute domains and illustration of their visual appearance.

| Attribute | Domain | Example | |
|---|---|---|---|
| Border Color | $\mathbb{N}^{3\times[0,255]}$ | gray | purple |
| Border Radius | $\mathbb{N}^{[0,20]} \cup \infty$ | 6 dp | 15 dp |
| Border Width | $\mathbb{N}^{[0,12]}$ | 0 dp | 4 dp |
| Main Color | $\mathbb{N}^{3\times[0,255]}$ | blue | purple |
| Padding | $\mathbb{N}^{[0,43]}$ | bottom | right |
| Shadow | $\mathbb{N}^{[0,12]}$ | 0 dp | 10 dp |
| Text Color | $\mathbb{N}^{3\times[0,255]}$ | gray | purple |
| Text Font Family | $\{\texttt{thin}, \texttt{light}, \texttt{regular}, \texttt{medium}, \texttt{bolt}\}$ | light | bold |
| Text Gravity | $\{\texttt{top}, \texttt{left}, \texttt{center}, \texttt{right}, \texttt{bottom}\}$ | left | right |
| Text Size | $\mathbb{N}^{[10,30]} \cup 0$ | 11pt | 14pt |
| Height | $\mathbb{N}^{[20,60]}$ | 30dp | 50dp |
| Width | $\mathbb{N}^{[25,275]}$ | 70dp | 110dp |

## B   REFINEMENT LOOP

**Stopping criterion**   For a given similarity metric (e.g., our learned attribute distance, pixel similarity, etc.) the stopping criterion of the refinement loop is defined using two hyperparameters:

- *early stopping*: stop if the similarity metric has not improved in the last $n$ iterations. In our experiments we use $n = 4$.
- *maximum number of iterations*: stop if the maximum number of iterations was reached. In our experiments we set maximum number of iterations to 8 when starting from the best prediction and to 100 if starting from random attribute values.

Table 4: Per attribute accuracy before and after applying the refinement loop when starting from the best predictions of the attribute prediction network for the real-world $\mathcal{D}_{\cong}^{gplay}$ dataset.

| | Accuracy on $\mathcal{D}_{\cong}^{gplay}$ | |
| --- | --- | --- |
| Attribute | Before Refinement | After Refinement |
| **Refined Attributes** | | |
| Text Size | 96.2% | 99.1% |
| Text Gravity | 94.4% | 96.3% |
| Text Font Family | 84.0% | 85.8% |
| Shadow | 93.7% | 97.3% |
| Width | 99.1% | 99.1% |
| Height | 93.6% | 97.3% |
| **Non-Refined Attributes** | | |
| Main Color | 99.1% | 99.1% |
| Text Color | 69.2% | 69.2% |
| Border Color | 92.6% | 92.6% |
| Padding | 81.1% | 81.1% |
| Border Width | 94.5% | 94.5% |
| Border Radius | 99.1% | 99.1% |
| All Attributes | 91.4% | 92.5% |

**Per-attribute accuracy** We provide per-attribute accuracy of the refinement loop in Table 4. As described in Section 5.2, we refined the following six attributes – text size, text gravity, text font, shadow, width and height. When considering only these six attributes, the refinement loop improves the accuracy by 2.3%, from 93.5% to 95.8%.

The remaining errors are mainly due to text color, padding and text font attributes. The padding achieves accuracy 81% and is difficult to learn as it often interferes with the text alignment and the same position of the text can be achieved with different attribute values. The text font attribute accuracy is 85% which is slightly improved by using the refinement loop but could be improved further. The worst attribute overall is the text color that achieves 69% accuracy. Upon closer inspection, the predictions of the text color are challenging due to the fact that the text is typically narrow and when rendered, it does not have a solid background. Instead, the color is an interpolation of the text color and background color as computed by the anti-aliasing in the rendering engine.

**Attribute dependencies** All the results presented in Table 4, as well as in our evaluation, are computed starting from the initial set of attribute values that typically contains mistakes. Since different attributes can (and do) dependent on each other, mispredicting one attribute can negatively affect the predictions of other attributes. As a concrete example, refining text font while assuming that all the other attribute values are correct leads to an improvement of 12% which is significantly higher than $\approx 2\%$ from Table 4.

To partially address this issue, the refinement loop uses a heuristic which ensures that all attributes have different values when used as input to the refinement loop. Concretely, if two attributes would have the same value, one of the values is temporarily changed to a random valid value and returned to the original value at the end of each refinement iteration.

## C  IMAGE PADDING

To improve robustness of our models on real-world images we experimented with three techniques of image padding shown in Figure 2. In *(a)* the image is padded with the edge values, in *(b)* the image is padded with a constant solid color and in *(c)* the image is simply extended to the required input size.

*(a)* Edge pixel padding     *(b)* Constant color padding     *(c)* Expanding bounding-box

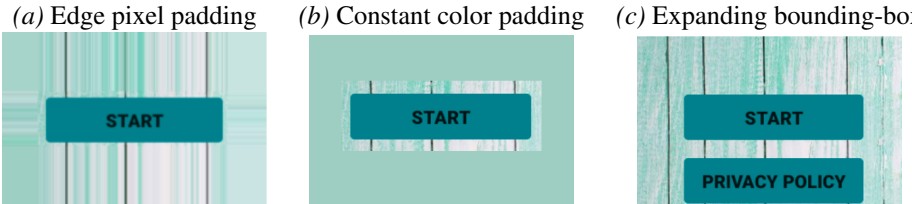

Figure 2: Illustration of different padding methods to resize the image to the network input size.

## D    COLOR CLIPPING USING SALIENCY MAPS

To improve color clipping results we are limiting the colors to which the predicted colors can be clipped by only considering the top 5 colors within the thresholded saliency map of the input image. An illustration of this process is shown in Figure 3, where *(a)* shows an initial input image, *(b)* the saliency map of the prediction, and *(c)* and *(d)* the thresholded saliency map (we use threshold 0.8) and the colors it contains.

*(a)* Input     *(b)* Saliency map     *(c)* Thresholded map     *(d)* Masked colors

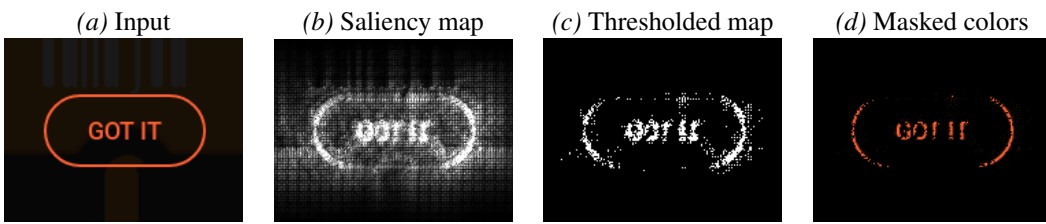

Figure 3: Restricting colors for color clipping.

## E    PERCEIVABLE ATTRIBUTE DIFFERENCE

We define the perceivable difference for each attribute in Table 5. We use $\epsilon$ to denote the distance between two attribute values. For all numerical attributes except colors, the distance is defined as the attribute value difference, i.e., $d(y_i, y_j) = y_i - y_j$. To better capture the difference between colors, we define their distance using the CIE76 formula (Schanda, 2007), denoted as dE. Furthermore, we provide illustration of the worse case perceivable difference for each attribute in Table 6.

Table 5: Perceivable difference definition for all attributes used in our work.

| Attribute | same ($=$) | similar ($\approx$) | different ($\neq$) |
|---|---|---|---|
| Border Color | $\epsilon \leq 5\text{dE}$ | $5\text{dE} < \epsilon \leq 10\text{dE}$ | $10\text{dE} < \epsilon$ |
| Border Radius | $\epsilon \leq 1\text{dp}$ | $1\text{dp} < \epsilon \leq 3\text{dp}$ | $3\text{dp} < \epsilon$ |
| Border Width | $\epsilon \leq 0\text{dp}$ | $0\text{dp} < \epsilon \leq 1\text{dp}$ | $1\text{dp} < \epsilon$ |
| Main Color | $\epsilon \leq 5\text{dE}$ | $5\text{dE} < \epsilon \leq 10\text{dE}$ | $10\text{dE} < \epsilon$ |
| Padding | $\epsilon \leq 1\text{dp}$ | $1\text{dp} < \epsilon \leq 3\text{dp}$ | $3\text{dp} < \epsilon$ |
| Shadow | $\epsilon \leq 0\text{dp}$ | $0\text{dp} < \epsilon \leq 2\text{dp}$ | $2\text{dp} < \epsilon$ |
| Text Color | $\epsilon \leq 5\text{dE}$ | $5\text{dE} < \epsilon \leq 10\text{dE}$ | $10\text{dE} < \epsilon$ |
| Text Font Family | same font | - | different font |
| Text Gravity | same gravity | - | different gravity |
| Text Size | $\epsilon \leq 1\text{sp}$ | $1\text{dp} < \epsilon \leq 2\text{dp}$ | $2\text{sp} < \epsilon$ |
| Height | $\epsilon \leq 1\text{dp}$ | $1\text{dp} < \epsilon \leq 3\text{dp}$ | $3\text{dp} < \epsilon$ |
| Width | $\epsilon \leq 2\text{dp}$ | $2\text{dp} < \epsilon \leq 4\text{dp}$ | $4\text{dp} < \epsilon$ |

Table 6: Examples of perceivable difference between two attribute values. For the same (=) and the similar (≈) perceivable difference, we include worst case examples.

| Attribute | Ground-truth | Examples of Perceivable Difference | | |
|---|---|---|---|---|
| | | same (=) | similar (≈) | different (≠) |
| Border Color | Ok | Ok | Ok | Ok |
| Border Radius | Ok | Ok | Ok | Ok |
| Border Width | Ok | Ok | Ok | Ok |
| Main Color | Ok | Ok | Ok | Ok |
| Padding | Ok | Ok | Ok | Ok |
| Shadow | Ok | Ok | Ok | Ok |
| Text Color | Ok | Ok | Ok | Ok |
| Text Font | Ok | Ok | - | Ok |
| Text Gravity | Ok | Ok | - | Ok |
| Text Size | Ok | Ok | Ok | Ok |
| Height | Ok | Ok | Ok | Ok |
| Width | Ok | Ok | Ok | Ok |

## F   DATASETS AND INFERRED IMPLEMENTATION VISUALIZATIONS

We provide illustrations of our approach for inferring Android `Button` implementations from images. Concretely, we include examples of images for which our approach works well, as well as examples where our models make mistakes. The visualizations for the synthetic $\mathcal{D}^{syn}$ and real-world $\mathcal{D}^{gplay}$ dataset of buttons found in Google Play Store applications are shown in Table 7 and Table 8, respectively. Each table row is divided into 4 parts: an image of the input, the preprocessed input image, a rendering of the predicted `Button` and a rendering of the refined `Button`.

Table 7: Visualization of the attribute predictions for the synthetic buttons in the $\mathcal{D}^{syn}$ dataset.

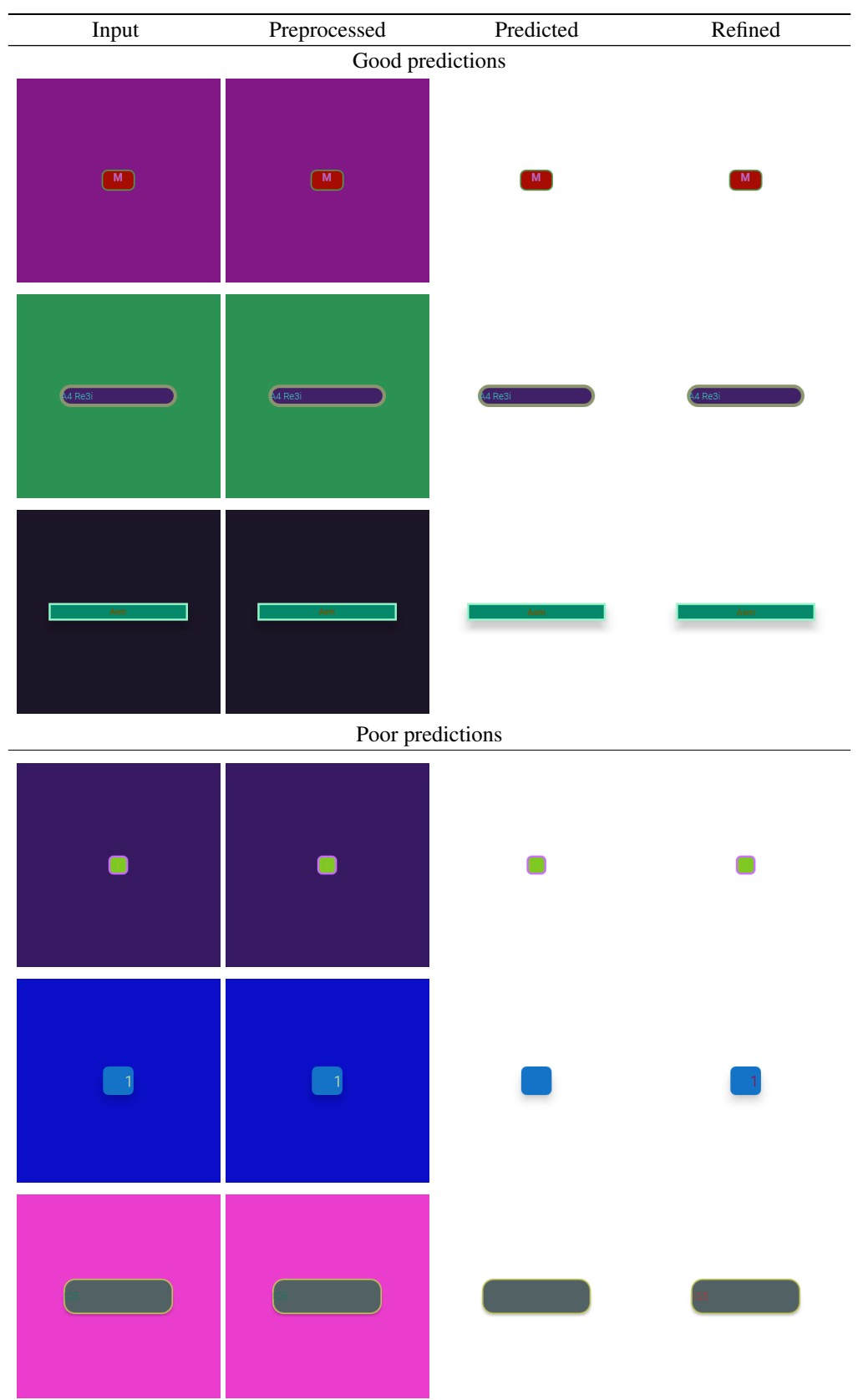

Table 8: Visualization of the attribute predictions for the real-world buttons in the $\mathcal{D}^{gplay}$ dataset.

| Input | Preprocessed | Predicted | Refined |
|:---:|:---:|:---:|:---:|
| Good predictions | | | |
|  |  |  |  |
|  |  |  |  |
|  |  |  |  |
| Poor predictions | | | |
|  |  |  |  |
|  |  |  |  |
|  |  |  |  |

