# OpenReview forum: "Learning to Infer User Interface Attributes from Images"
_ICLR.cc/2020/Conference — Reject_

### Official Review · AnonReviewer3 · 2019-10-20
**Official Blind Review #3**

**Rating:** 1

**Review:**

Authors proposed an algorithm to predict the attribute of GUI elements from rasterized design images. The problem is separated into two steps. The first step is to predict initial values of the attributes (border width, color, padding etc) from the image where the type of UI element and set of attributes are already known. Authors designed a typical convolutional DNN for each of the attributes. The second step is to learn a policy \pi to iteratively adjust one attribute a time until the final rendering matches input image pixel-perfect.

Authors conducted the experiment with large synthetic data set of Android buttons, and evaluate the performance with a held-out synthetic set as well as a 110 hand crafted real world buttons set sampled from apps in Google Play App Store. Several variations of the same model were compared. The result showed that network structure, padding strategy (this is a bit unexpected), background type and color selection strategy all affect the accuracy significantly.

Reviewer has concern about the scope and application value of the problem as a research paper. A number of key prior assumptions have to be made to let the algorithm work: the type of the UI element need to be known; the list of attributes and their value ranges need to be fixed beforehand and each of the attribute demands a DNN; the refinement iteration has the actual rendering in the loop which could be costly on current App development platforms.

Feedback questions.

1) The abstract mentioned vector image as input but the main body only discussed rasterized images.

2) Since the rendering process could be costly, it's useful to discuss the speed of convergence in the attribute value adjustment iterations.

3) Reviewer is interested in the nature of the error (7.5% loss) but it's not discussed.

4) In related work, authors mentioned the REINFORCE algorithm by Williams et al 1992 is expensive. It could help the reader if a brief explanation of why it's expensive is provided.

5) In Section 3 Background, authors mentioned that the attributes are not independent of each other, which is a major challenge. Reviewer would like to see some discussion or experiment data on how this affects the process and how did the current algorithm address it.

6) It's a bit surprise that color clipping method has a big impact on accuracy. Some examples could have helped the reviewer understand it.

7) In Section 4.2 first paragraph, it seems that the user of the algorithm need to set the [-c, c] clipping values manually per feature. This sounds like quite some prior knowledge and hand-tuning.

8) In Section 4.2. Second paragraph, Reviewer can understand the nessesity of additive and subtractive operations, but why multiplication?

9) In the equation to the end of page 5, do we need an extra outer bracket for the denominator? By the way, the equations should be numbered for easier reference.

10) The task of predicting Android button attributes, while practical, seems over-simplified. Reviewer suggests at least experiment with a set of common UI elements to proof the horizontal performance.

11) In Section 5.1, Reviewer respect the experiment results but doesn't understand why solid color background provides the best variety but screenshots don't. May need more analysis and explanation.

12) In Table 1, the first line for variant (C) also looks pretty good, or even better than core on the Android app store dataset.

13) In Section 5.1, the effect of color clipping selection seems very specific to applications with a fixed color palette. While this is indeed the majority, this prior knowledge need to be speciifed clearly by saying it's tailored towards such applications (or use more examples to proof that's not the case).

14) In Table 2: Pixel Sim's performance on Best Prediction Initialization seems pretty good, and Reviewer believes this is the more practical scenario. Is a more complicated Siamese Network justified?



**Experience Assessment:**

I do not know much about this area.

**Review Assessment: Checking Correctness Of Derivations And Theory:**

I carefully checked the derivations and theory.

**Review Assessment: Checking Correctness Of Experiments:**

I carefully checked the experiments.

**Review Assessment: Thoroughness In Paper Reading:**

I read the paper thoroughly.

---

> ### Author Response · Authors · 2019-11-09
> **Response to Reviewer #3 (part 1)**
>
> We thank the reviewer for the thorough comments.
>
> We would like to clarify that the main scope of our work is to explore a new domain of learning to infer user interface attributes, the challenges it contains and experimentally showing how they can be addressed for a non-trivial set of attributes (including comprehensive evaluation of different design decisions, network architectures and various optimizations used throughout our work). To achieve this we have selected Android button component as: (i) it is the most common component used by existing applications, and (ii) provides high variety in the attributes (e.g., both categorical and continuous, colors, text attributes and visual attributes such as border and shadows). To our best knowledge we are the first work to explore this domain and we will release or source code, datasets as well as the learning infrastructure to support further research in this domain.
>
> Please find the answers to your questions below:
>
> Q: A number of key prior assumptions have to be made to let the algorithm work: 1) the type of the UI element need to be known; 2) the list of attributes and their value ranges need to be fixed beforehand; 3) each of the attribute demands a DNN; 4) the refinement iteration has the actual rendering in the loop which could be costly on current App development platforms.
>
> A: We disagree.
> 1) Although the type of the UI element needs to be known, the list of UI types is easy to obtain either by looking at the documentation of the rendering engine or by downloading existing application and inspecting which components they use.
>
> 2) To obtain value ranges, we can collect a set of existing applications and compute statistics about which attributes and value ranges they use. In fact, this is exactly what we did in our work and the reason why we decided to use the button component (its the most widely used component with a wide variety of attributes).
>
> 3) It is also possible to train a DNN that is shared across multiple attributes. We experimented with such a design and it achieved comparable results to training the networks separately. The reason why we chose to train the networks individually is because we believe this is a more modular approach. In particular, adding a new attribute requires training only a single DNN (specialized for that attribute) rather than retraining the whole system.
>
> 4) While it is true that the rendering loop can be costly, the combination of attribute prediction and refinement requires only up to 5 refinement iterations which take less than a second in our experiments.
>
> Q: The abstract mentioned vector image as input but the main body only discussed rasterized images.
>
> A: Correct, we consider rasterized images since this is a more general setting and allows us to infer attributes for arbitrary images regardless of how they were created. For example, even though some images in our test dataset contain screenshots of web elements (since the application was showing a web page), we were still able to infer the corresponding attributes of a native Android component that looks visually similar. However, it would also be possible to include the metadata extracted from the vector image as an additional input (alongside of the rasterized image) which will most likely lead to a further accuracy improvement.
>
> Q: In Table 2: Pixel Sim's performance on Best Prediction Initialization seems pretty good, and Reviewer believes this is the more practical scenario. Is a more complicated Siamese Network justified?
>
> A: Quite the opposite. What Table 2 shows is that when starting from the best prediction initialization, our learned distance function is _the only_ model that achieves an improvement. All the other distance function either worsen the results or in the best case, do not change anything. In other words, doing nothing is strictly better than using any of the baseline distance functions. The difference between our learned distance function and the baselines can be more clearly seen when starting from the random initial attribute values. Here, our method is the only one that successfully refined the attribute values.
>
> Q: Since the rendering process could be costly, what is the speed of convergence in the attribute value adjustment iterations?
>
> A: The convergence is fast and converges on average after 4-5 iterations (when starting from the predictions computed by the attribute network). This is partially because the starting predictions are already good and the refined values are selected by sampling from the learned distribution of the most likely miss-predicted values (rather than picking them at random).

---

> ### Author Response · Authors · 2019-11-09
> **Response to Reviewer #3 (part 2)**
>
> Q: Authors mentioned the REINFORCE algorithm by Williams et al 1992 is expensive. It could help the reader if a brief explanation of why it's expensive is provided.
>
> A: The main reason is that it is extremely sample inefficient and each sample needs to be obtained by running the policy in an environment (in our case rendering an image). This is typically orders of magnitude slower that running a training with a fixed dataset (e.g., as done in imitation learning). A more technical explanation with a possible improvements is provided in a recent work of Botvinick et. al. Reinforcement Learning, Fast and Slow. Trends Cogn Sci. 2019 May;23(5):408-422.
>
> Q: Could you discuss the nature of the error (7.5% loss)?
>
> A: The loss is mainly due to text color, padding and text font attributes. The padding achieves accuracy 81% and is difficult to learn as it often interferes with the text alignment and the same position of the text can be achieved with different attribute values.  The text font attribute accuracy is 85% which is already improved by using the refinement loop but could be improved further. The worst attribute overall is the text color that achieves 69% accuracy. Upon closer inspection, the predictions of the text color are challenging due to the fact that the text is typically narrow and when rendered, it does not have a solid background. Instead, the color is an interpolation of the text color and background color as computed by the anti-aliasing in the rendering engine.
>
> Q: Authors mentioned that the attributes are not independent of each other, which is a major challenge. Could you discussion or provide experimental data on how this affects the process and how did the current algorithm address it.
>
> A: A typical example of conflicting attributes are any color attributes that are rendered next to each other in the image. For example, if the border color and background color are the same, then increasing the border width and increasing the size of the component has the same visual effect. Another example is a combination of border width, radius and the component size. Here, different combinations of these attributes can lead to border radius that is visually the same.
>
> To address this issue in the refinement loop we use a simple heuristic which ensures that all attributes have different values when used as input to the refinement loop. Concretely, if two attributes would have the same value, one of the values will be temporarily changed and returned to the original value at the end of each refinement iteration.
>
> Q: It's a bit surprising that color clipping method has a big impact on accuracy. Some examples could have helped the reviewer understand it.
>
> A: Intuitively, as there can be many colors in the image, it can be difficult for the network to learn which ones are relevant for the given attribute. The color clipping helps the network to remove many of the irrelevant colors and reduces the task of predicting among all colors to an easier task of selecting only among small set of colors. We  provide a visualization of this process in the Appendix C.
>
> Q: It seems that the user of the algorithm need to set the [-c, c] clipping values manually per feature. This sounds like quite some prior knowledge and hand-tuning.
>
> A: Quite the opposite, these are hyperparameters and do not require any prior knowledge or hand-tuning. In fact, in our experiment we did not perform any tuning as use c=5 for all comparable attributes.
>
> Q: In Section 4.2. Second paragraph, Reviewer can understand the necessity of additive and subtractive operations, but why multiplication?
>
> A: Overall, adding different combinations helped to improve the network accuracy. A possible intuition behind the multiplication is that it allows the network better capture the magnitude of the difference between the features.

---

> ### Author Response · Authors · 2019-11-09
> **Response to Reviewer #3 (part 3)**
>
> Q: In the equation to the end of page 5, do we need an extra outer bracket for the denominator?
>
> A: Correct, thanks for pointing this out. The summation in the denominator is over both ($1-\pi$) and not just 1. We will update this in our paper (along with numbering the equations)
>
> Q: Why does the solid color background provide the best variety but screenshots don't?
>
> A: The main reason for this are twofold. First, inserting the components into existing screenshots is non-trivial and often introduces visual artefacts that are exploited by the network during the training. As there artefacts are not present during testing (there each component already contains the background provided by the application) the performance decreases. Secondly, cropping and padding techniques allows us to remove most of the complexity found in screenshots by focusing only at small area around the component which is typically quite regular and similar to a solid background.
>
> Q: In Table 1, the first line for variant (C) also looks pretty good, or even better than core on the Android app store dataset.
>
> A: Table 1 shows how different design decisions proposed in our work affect the performance with the goal of finding the best configuration. As a result, it is expected (and desirable) that other configurations might lead to better results. The purpose of the ‘core’ configuration was to provide a strong model against which all the other experiments can be compared to.
>
> Q: The effect of color clipping selection seems very specific to applications with a fixed color palette. While this is indeed the majority, this prior knowledge need to be specified clearly by saying it's tailored towards such applications
>
> A: Indeed, color clipping works best when the application uses a fixed color palette. We will clarify this point in our paper.

---

> ### Comment · AnonReviewer3 · 2019-11-11
> **Response to Authors' response.**
>
> First I want to thank the authors for the detailed answers. Most of the questions are answered well, I hope authors can make them clear in the paper too.
>
> Here are the two questions I'm still not 100% convinced.
>
> Question 10: I understand that the Button is a widely, probably mostly used UI element, and choosing it as the object to study makes sense.
>
> However,  the authors claim that this work is about generic UI elements in the title, abstract and the second paragraph in Introduction. I still think we need more cases than Android Button to prove that the method works for generic UI elements, which is a quite diverse set of things.
>
>
> Question 14: Authors argued that the Siamese network is the only similarity function that brings an improvement, which I agree.
>
> My concern has been the cost/benefit ratio: Siamese network is significantly more complicated than PixelSim (or doing nothing) but only brings marginal improvements over best prediction. We may need more evidence to show it's necessity. For example, if somehow the experiments on other UI elements showed strong improvements over the baseline.
>
>
>
> Last note to the Area Chair: I'm not working in the field of UI pixel-to-code generation. All my comments are made with my experience in generic ML research (mostly NLP and Data Mining on the Web) and real-world mobile apps development. It could help if at least one of the reviewers has research background on this matter.

---

> > ### Author Response · Authors · 2019-11-12
> > **Overhead of the refinement loop**
> >
> > Question: My concern has been the cost/benefit ratio: Siamese network is significantly more complicated than PixelSim (or doing nothing) but only brings marginal improvements over best prediction. We may need more evidence to show it's necessity. For example, if somehow the experiments on other UI elements showed strong improvements over the baseline.
> >
> > Answer: We agree that to show that the refinement loop is necessary it would require more evidence. The only claim we can currently support is that the refinement loop does lead to an improvement, as shown by our experiments.
> >
> > However, we would like to point out that the Siamese network design as well as our choice of imitation learning are made such that they incur only small overheard, both for training and the required infrastructure. This is because:
> >
> > (i) the Siamese Network reuses the already trained attribute prediction networks by combining their learned latent features (from the second to last layer) and adds learnable transformation on top of them. As discussed in our evaluation, we initialize the Siamese networks with the pretrained weights of the best attribute prediction network.
> >
> > (ii) our approach is based on generating synthetic datasets, therefore the infrastructure required to render attributes is reused.
> >
> > (iii) the choice of imitation learning means that training a policy is phrased as a sequence of supervised learning tasks. This again reuses the infrastructure and training methods used for training the attribute prediction network. This is in contrast to using other reinforcement learning methods (e.g., REINFORCE) which are more difficult to train and require specialized training algorithms and infrastructure for scalable training.
> >
> > The main source of overhead comes from selecting which attribute to refine (and to which value) and determining how many iterations to perform (i.e., when to stop).

---

### Official Review · AnonReviewer1 · 2019-10-21
**Official Blind Review #1**

**Rating:** 3

**Review:**

This paper proposes an approach for reverse-engineering webpages using Siamese networks and imitation learning. While the idea of using synthetic data (which can be easily procedurally generated) to do this reverse-engineer training is very clever, prior work has exploited it also. Novel elements include the attribute refinement using imitation learning, and the authors show the effect of this step, but the improvement is small. Thus, the limited novelty and not very convincing results make the question the potential impact of this paper.

Some questions:
a) The authors mention they cannot use a GAN-style method because all generated images are by definition true/real; how about learning whether a *pair* is real or fake? (where the pair consists of the design specification and the rendered version).
b) Are the baselines strong enough? None of them seem to be from recent prior work. How about a direct comparison to some of the work listed in the second para on page 2?

**Experience Assessment:**

I do not know much about this area.

**Review Assessment: Checking Correctness Of Derivations And Theory:**

I assessed the sensibility of the derivations and theory.

**Review Assessment: Checking Correctness Of Experiments:**

I assessed the sensibility of the experiments.

**Review Assessment: Thoroughness In Paper Reading:**

I made a quick assessment of this paper.

---

> ### Author Response · Authors · 2019-11-09
> **Response to Reviewer #1**
>
> We thank the reviewer for the comments and clarifying questions. We provide detailed answers below:
>
> Q: The attribute refinement using imitation learning is novel but the improvement seem small.
>
> A: The improvements are significant for some attributes such as text font where the accuracy increases by ~12% or text gravity where the improvement is from 96.2% to almost perfect 99.5%. The overall improvement shown in Table 2 is lower since we average over multiple attributes, many of which already achieve very high accuracy. The majority of remaining mistakes are due to color mispredictions which is partially addressed by introducing optimizations to our architecture that are specific for this attribute.
>
> Q: The authors mention they cannot use a GAN-style method because all generated images are by definition true/real; how about learning whether a *pair* is real or fake? (where the pair consists of the design specification and the rendered version).
>
> A: Such design would roughly correspond to our attribute refinement loop. Here we train a discriminative model to learn a suitable distance metric in the attribute space. In contrast, in GANs, learning a distance between two images is only a by-product of training a generative model capable of modifying one of the input images such that they look more similar. Training such a generative network is not only much more challenging but also redundant given our goal is to recover the correct attributes (that are passed to a rendering engine) rather than recover the correct image (what the GAN is trained to do).
>
> However, we note that there are challenges in our work where it would be interesting to try and apply GANs. For example, one of the challenges in generating a synthetic dataset is how to insert a component into an existing application screenshot without introducing visual artifacts. Here, one could try to train a GAN to insert the component in such a way that it cannot be easily distinguished that it does not belong in the image with the additional constraint that the attributes of the component need to be preserved. However, the additional constraint that the attributes need to be preserved is non-trivial and an interesting research question on its own.
>
> Q: Are the baselines strong enough? None of them seem to be from recent prior work. How about a direct comparison to some of the work listed in the second para on page 2?
>
> A: The reason why we do not provide experimental comparison to the prior work (e.g., second paragraph on page 2) is because such comparison is unfortunately not possible. Even though the high level task is the same, inverting rendering engines to interpret images, the actual datasets and network architectures are specialized for the given domain. For example, it makes little sense to use an architecture specialized to predict camera angle and instead try to predict border width. To our best knowledge, there is no prior work that we can compare to in the same domain as we are the first work to explore solving the task of learning to inferring user interface attributes.

---

### Official Review · AnonReviewer2 · 2019-11-05
**Official Blind Review #2**

**Rating:** 8

**Review:**

The paper proposes an approach to infer the attribute values of an input image representing a user interface. The model first infers the most likely initial attribute values, and iteratively refine them to improve the similarity between the input image and the interface generated from the newly inferred attributes. The model is trained on synthetic datasets generated by a black box rendering engine, and generalizes well to real-world datasets. To address the issues of pixel based metrics and mean squared error, the authors instead uses the probability that two images are equal in the attribute space to define the cost between these two images.

Although I'm not familiar with the problem addressed by the paper, I found the paper very clear and well written. Overall, the method is sensible and elegant, and could easily be applied to other domains. My only disappointment is maybe the fact that only the Android Button was considered, and it is not clear how the model would perform with other and more sophisticated Android components.

A few questions for the authors:
- How many steps do you perform in the refinement loop? This is an important information, but I couldn't find it in the paper. Typically, I was surprised to see in the first row of Table 2 that the model with a random attribute initialization can reach such a high performance. But I imagine that you need many more iterations to converge if you start from random attributes than from the best prediction initialization?
- Also, what is the stopping criterion? Do you decide to stop when none of the proposed attribute changes improve the pixel-level accuracy?

**Experience Assessment:**

I do not know much about this area.

**Review Assessment: Checking Correctness Of Derivations And Theory:**

I assessed the sensibility of the derivations and theory.

**Review Assessment: Checking Correctness Of Experiments:**

I assessed the sensibility of the experiments.

**Review Assessment: Thoroughness In Paper Reading:**

I read the paper at least twice and used my best judgement in assessing the paper.

---

> ### Author Response · Authors · 2019-11-09
> **Response to Reviewer #2**
>
> We thank the reviewer for the comments and clarifying questions. We provide detailed answers below:
>
> Q: How many steps do you perform in the refinement loop? I imagine that you need many more iterations to converge if you start from random attributes than from the best prediction initialization?
>
> A: Correct, when starting from random attributes we need many more iterations to converge. Concretely, we perform 100 refinement iterations when starting from random attribute values whereas we perform maximum of 8 when initializing with the best prediction.
>
> Q: What is the stopping criterion? Do you decide to stop when none of the proposed attribute changes improve the pixel-level accuracy?
>
> Not quite. As can be seen in our experiments, the pixel-level accuracy is not a good measure for comparing the similarity of images in their attribute space. Therefore, instead of using pixel-level accuracy, we use our learned attribute distance to decide when to stop.
>
> Concretely, for a given similarity metric (e.g., our learned attribute distance, pixel similarity, structural similarity or wasserstein distance) the stopping criterion is defined using two hyper parameters:
>   - (early stopping) stop if the similarity metric has not improved in the last $n$ iterations (in our experiments $n=4$)
>   - (maximum number of iterations) stop if the maximum number of iterations was reached (i.e., 8 when starting from the best prediction, 100 if starting from random attribute values).
>
> Q: I was surprised to see in the first row of Table 2 that the model with a random attribute initialization can reach such a high performance.
>
> A: The fact that the accuracy of our refinement loop is almost the same regardless of the starting attributes can be seen as a sanity check. It shows that, given enough iterations, the learned neural policy as well as the overall refinement loop works well and does not get stuck in a local minima (e.g., by refining conflicting set of attribute values). It also shows that even though the baseline similarity metrics use an idealized setting (i.e., the input images are perfectly aligned which significantly improves their accuracy), they all get stuck at different points of the refinement.
>
> Q: My only disappointment is maybe the fact that only the Android Button was considered, and it is not clear how the model would perform with other and more sophisticated Android components.
>
> A: Since each component consists of a set of attributes, our main focus was to design and evaluate our approach on a wide range of attributes. For this reason we have selected Android button component as: (i) it is the most common component used by existing applications, and (ii) provides high variety in the attributes (e.g., both categorical and continuous, colors, text attributes and visual attributes such as border and shadows).
>
> Having said that, we do agree that experimenting with other components would make our work stronger and provide experimental support that indeed our technique scales well.

---

### Author Response · Authors · 2019-11-12
**Paper Revision**

Dear reviewers,

We have updated our paper based on your comments and questions. The main changes include:

1) [Abstract/Introduction]: We clarify that the main scope of our work is to explore a new domain of learning to infer user interface , the challenges it contains and experimentally showing how they can be addressed for a non-trivial set of attributes. We explicitly say both in abstract and introduction that our is evaluated on Android button component. The motivation behind this choice was that: (i) it is the most common component used by existing applications, and (ii) provides high variety in the attributes (e.g., both categorical and continuous, colors, text attributes and visual attributes such as border and shadows).

2) [Abstract] remove mentioning vector image to avoid confusion as the input to our approach is rasterized image

3) [Section 4.2] Clarify the usage of [-c, c]

4) [Evaluation] Clarify that the color clipping is designed for solid color palettes and not for gradient colors

5) [Appendix B] Provide details of the stopping criterion used in the refinement loop

6) [Appendix B] Discuss the nature of errors and and provide per-attribute accuracy breakdown of the refinement loop

We believe that our work is a useful step in developing practical tools that support a wide range of different attributes and components, beyond those considered in our work. We will release or source code, datasets as well as the learning infrastructure to support further research in this domain.

---

### Decision · Program_Chairs · 2019-12-19

**Decision:**

Reject

**Comment:**

The majority of reviewers suggest rejection, pointing to concerns about design and novelty. Perhaps the most concerning part to me was the consistent lack of expertise in the applied area. This could be random bad luck draw of reviewers, but more likely the paper is not positioned well in the ICLR literature. This means that either it was submitted to the wrong venue, or that the exposition needs to be improved so that the paper is approachable by a larger part of the ICLR community. Since this is not currently true, I suggest that the authors work on a revision.